# Usefulness of the Inferior Articular Process’s Cross-Sectional Area as a Morphological Parameter for Predicting Central Lumbar Spinal Stenosis

**DOI:** 10.3390/jcm9010214

**Published:** 2020-01-13

**Authors:** Sooho Lee, Taeha Lim, Young-Seob Lim, Young Uk Kim

**Affiliations:** 1Department of Anesthesiology and Pain Medicine, Asan Medical Center, University of Ulsan College of Medicine, Seoul 05505, Korea; rysooho84@gmail.com; 2Department of Anesthesiology and Pain Medicine, Eulji General Hospital, Eulji University College of Medicine, Seoul 01830, Korea; limtaeha@hanmail.net (T.L.); lucete2012@gmail.com (Y.-S.L.); 3Department of Anesthesiology and Pain Medicine, Catholic Kwandong University, College of Medicine, International ST. Mary’s Hospital, Incheon 22711, Korea

**Keywords:** inferior articular process, cross-sectional area, spinal stenosis

## Abstract

Hypertrophy of facet joints is associated with a high risk of central lumbar spinal stenosis (CLSS). However, no research has reported the effect of inferior articular process hypertrophy in CLSS. We hypothesize that the inferior articular process’s cross-sectional area (IAPCSA) is larger in patients with CLSS compared to those without CLSS. Data on IAPCSA were obtained from 116 patients with CLSS. A total of 102 control subjects underwent lumbar spine magnetic resonance imaging (LS-MRI) as part of a routine medical examination. Axial T1-weighted images were obtained from the two groups. Using an imaging analysis system, we investigated the cross-sectional area of the inferior articular process. The average IAPCSA was 70.97 ± 13.02 mm^2^ in control subjects and 88.77 ± 18.52 mm^2^ in patients with CLSS. CLSS subjects had significantly greater levels of IAPCSA (*p* < 0.001) than controls. A receiver operating characteristic (ROC) curve was plotted to determine the validity of IAPCSA as a predictor of CLSS. The most suitable cut-off point of IAPCSA for predicting CLSS was 75.88 mm^2^, with a sensitivity of 71.6%, a specificity of 68.6%, and an area under the curve (AUC) of 0.78 (95% CI: 0.72–0.84). Greater IAPCSA levels were associated with a higher incidence of CLSS. These results demonstrate that IAPCSA is a useful morphological predictor in the evaluation of CLSS.

## 1. Introduction

Central lumbar spinal stenosis (CLSS) is the most common spinal disorder in patients aged 60 years and above [1]. The clinical manifestation of CLSS is chronic pain with a variable clinical manifestation that affects normal work and daily life and limits activity [2]. It also presents with buttock pain, neurogenic intermittent claudication, and motor and sensory disturbances in the leg [2,3,4,5]. The course of CLSS is slow and insidious. The initial pain is not severe. It is combined with muscular discomfort that can be improved by rest or changing position [1,5]. Final pathologies underlying pain in CLSS patients are neurological damage to the spinal nerve compression and cauda equina [6]. CLSS is characterized by lumbar spinal canal narrowing that is caused by proliferation of the superior articular process, mechanical compression of the spinal nerve roots, hyperostosis of the vertebral posterior border, a hypertrophied ligamentum flavum, and protrusion of the intervertebral disc [7]. Previous research has demonstrated that morphological features, including the superior articular process’s cross-sectional area (CSA), the spinal canal’s CSA, the dural sac’s CSA, and the ligamentum flavum’s thickness are associated with greater incidences of CLSS [8,9,10,11]. To date, however, no study has reported the effect of inferior articular process (IAP) hypertrophy on CLSS. IAP is a process of the vertebra that lies on each side of the neural arch, projects downward, and articulates with a superior articular process of the next more caudal vertebra. We hypothesized that the CSA of IAP is an adjuvant morphological feature in the diagnosis of CLSS. To analyze the connection between CLSS and hypertrophied IAP, we investigated the inferior articular process’s cross-sectional area (IAPCSA). We used lumbar spine magnetic resonance imaging (LS-MRI) to compare the IAPCSA between normal subjects and CLSS patients.

## 2. Methods and Materials

### 2.1. Patients

This research was reviewed and approved by the Institutional Review Board (IRB) of the International St. Mary’s Hospital, Catholic Kwandong University, College of Medicine, Incheon (IRB number: IS18RISI0046). Patient medical charts from the period between June 2017 and April 2019 were acquired from the musculoskeletal imaging system. Patients above the age of 50 were included if they had clinical symptoms compatible with CLSS (such as neurogenic intermittent claudication and disturbances in the lower leg), at the most stenotic level (L4/5), and LS-MRI within 3 months of diagnosis. Exclusion criteria were: presence of history of spinal injury, a congenital spine defect, previous lumbar surgery, or an osteoporotic vertebral lumbar fracture.

A total of 116 patients had a diagnosis of CLSS, which was confirmed by two board-certified, experienced musculoskeletal radiologists, including 38 (32.7%) men and 78 (67.3%) women with an average age of 63.14 ± 8.41 years (range: 50–82 years). The IAPCSA of subjects with or without CLSS was compared with that of control subjects who had undergone LS-MRI as part of a non-symptomatic medical checkup. Patients in the control group had no CLSS-related symptoms. The group of control subjects consisted of 102 patients (22 males and 80 females) with an average age of 62.25 ± 8.51 years (range: 50–83 years). The IAPCSA in the group of control subjects was also examined at the L4/5 facet joint level (Table 1).

### 2.2. Imaging Parameters

LS-MRI was performed on a 3T (Avanto, Siemens Medical Solution, Germany) with 3T-MRI scanners (MAGNETOM Skyra, Siemens Healthcare, Erlangen, Germany). The LS-MRI examination was conducted and T1-weighted axial turbo spin echo (TSE) images were acquired at a thickness of <4 mm, a repetition time of 700 ms, an echo time of 12 ms, a 180 × 180 field of view, and a 384 × 346 matrix. LS-MRI data were transferred to an INFINITT system (INFINITT picture archival and communications system (PACS system), Seoul, Korea).

### 2.3. Image Measurement

T1-weighted TSE axial LS-MR images were obtained at the L4/5 facet joint level. A PACS system was used to measure the IAPCSA at the L4/5 facet joint level in the LS-MRI. The IAPCSA was measured as the outline of the inferior articular facet’s anterolateral surface at the L4/5 level (Figure 1).

### 2.4. Statistical Analysis

Statistical analysis was carried out using IBM SPSS version 22 for Windows (IBM Corporation, Armonk, NY, USA). Statistical algorithms were used to calculate SDs. The *p*-values of less than 0.05 were considered statistically significant. An unpaired *t*-test was used to compare the IAPCSA between CLSS and control groups. One-way ANOVA was used to analyze the correlation between age-related changes and IAPCSA. The validity of IAPCSA for diagnosis of CLSS was determined using a receiver operating characteristic (ROC) curve analysis.

## 3. Results

Demographic data were not significantly different between the control and CLSS groups (Table 1). The mean IAPCSA was 70.97 ± 13.02 mm^2^ for the control group and 88.77 ± 18.52 mm^2^ for the CLSS group. CLSS patients had a significantly (*p* < 0.05) higher IAPCSA than the control group (Table 1). The average IAPCSA of control subjects was 69.40 ± 12.31 mm^2^ in those aged 50–59 years, 70.28 ± 13.39 mm^2^ in those aged 60–69 years, and 75.81 ± 13.43 mm^2^ in those aged 70–82 years. There was no significant difference in IAPCSA among different age groups of control subjects based on one-way ANOVA (F = 1.79; df = 2; *p* = 0.172) (Table 2). The mean IAPCSA of CLSS subjects was 91.38 ± 21.36 mm^2^ for those aged 50–59 years, 85.93 ± 17.09 mm^2^ for those aged 60–69 years, and 89.14 ± 15.65 mm^2^ for those aged 70–83 years. There was no significant difference in IAPCSA among different age groups of CLSS subjects (F = 0.959; df = 2; *p* = 0.386) (Table 3). The ROC curve analysis was performed to determine the validity of the IAPCSA as a predictor of CLSS. The most suitable cut off-point of IAPCSA for predicting CLSS was 75.88 mm^2^, with a sensitivity of 71.6%, a specificity of 68.6%, and an area under the curve (AUC) of 0.78 (95% CI: 0.72–0.84) (Table 4 and Figure 2).

## 4. Discussion

The aim of this research was to investigate the relationship between CLSS and IAPCSA using LS-MRI. We demonstrated a positive correlation between CLSS and IAPCSA. We found that the cut-off point of IAPCSA (75.88 mm^2^) had a sensitivity of 71.6%, a specificity of 68.6%, and an AUC of 0.78 to predict CLSS. Our results suggest that IAPCSA is an accurate and objective morphological parameter for CLSS prediction. CLSS is defined as spinal canal narrowing secondary to ligamentum flavum hypertrophy, with facet joint arthritic changes, disc bulging, protrusion, and extrusion, combined with osteophytes, resulting in signs and symptoms caused by compression and entrapment of the intraspinal, nervous, and vascular structures [3,12,13,14]. Prevalence of CLSS is approximately 30% in the elderly. Symptoms of CLSS are related to a neurovascular pathomechanism such as venous congestion, arterial flow in the cauda equina, nerve root excitation by inflammation, increased epidural pressure, or direct compression in the spinal canal [15,16]. Previous investigators have reported associations of the superior articular process’s CSA, the dural sac’s CSA, the spinal canal’s diameter, and the ligamentum flavum in LS-MRI with symptoms and signs of CLSS. Lim et al. have found that higher superior articular process CSA values are associated with a higher probability of CLSS. In this investigation, the best cut-off point of the superior articular process’s area was 112.12 mm^2^ with a sensitivity of 84.4% and a specificity of 83.9%. This value is higher than that of the IAPCSA. The cause of the difference between these two values is presumably that the CSA of the superior articular process is larger than that of the IAP [10]. Ogikubo et al. have reported a significant relationship between shorter walking distances and a smaller dural sac CSA [11]. Kim et al. have demonstrated that a longer walking distance is associated with a larger dural sac CSA [9]. Korse et al. have found that patients with cauda equina syndrome due to lumbar disc herniation have significant smaller anteroposterior spinal canal diameters in the lumbar lesion than patients with lumbar disc herniation without cauda equina syndrome [17]. Yoshiiwa et al. have reported that hypertrophy of ligamentum flavum development is associated with severe disc degeneration, segmental instability, and severe facet joint osteoarthritis [18]. However, no study has reported the correlation of CLSS and hypertrophy of IAP as a morphological parameter in lumbar spine MRI. IAPs are suspended from the lamina on each side. The superolateral portion of the lamina can be perceived as lying between the IAP and the superior articular process of the parent vertebra. IAPs of lumbar vertebrae are located on the anterior and lateral convex. They cover their anterolateral surface, which possesses inferior articular facets [19]. Thus, we hypothesized that the hypertrophy of IAP might have a close relation to facet joint arthritis and CLSS. Our results demonstrated an association between IAPCSA and CLSS. The positive association between IAPCSA and CLSS can be explained by an increase in IAPCSA correlated with increased risk of CLSS. In our research, the optimal cut-off point of the IAPCSA was 75.88 mm^2^ with a sensitivity of 71.6%, a specificity of 68.6%, and an AUC of 0.78. Our interpretation of such an association is based on our understanding that the IAP hypertrophy process begins with mechanical friction during lumbar rotation, extension, and flexion. Because the inferior articular facet meets the superior articular facet at the posterior and medial concavity of the adjoining surface [19], these frictions can exert continuous pressure on zygapophyseal joints and alter morphological features of IAP, leading to a high degree of abrasion [20]. Burke et al. have reported a case of spinal cord trauma caused by direct transmission of force from an ostectomy to a symptomatic ossified ligamentum flavum during resection of an IAP [21]. Das et al. have reported that an asymmetrical hypertrophy of IAP might be related to disc degeneration and CLSS, and that it may be a cause of buttock and low back pain [22]. Our results suggest that IAPCSA is a new adjuvant morphological parameter for CLSS prediction. Our study included individuals above the age of 50. This is because only minimal cartilage changes are correlated with facet degeneration in those aged younger than 50, and that the hypertrophic change in osteoarthritis advances with the aging process [10]. In this research, we analyzed the IAPCSA from LS-MRI images and visualized IAP in transverse T1-weighted LS-MRI. LS-MRI provides a high-quality view of the IAP and IAP hypertrophic changes [10,21]. Our research had a few weaknesses. First, we had a relatively small sample size. Future studies enrolling a large population are needed to provide more persuasive and typical morphological results. Second, there might be measurement errors associated with the analysis of IAPCSA in LS-MRI. We tried to measure the IAPCSA in transverse T1-weighted images at the level of the L4/5 facet joint. However, transverse images can be inhomogeneous due to different cutting angles of the LS-MRI resulting from technical deficiencies and individual anatomic variations. A 4 mm slice of a transverse T1-weighted LS-MR image is thicker than an ideal image. Third, several different methods are known to discriminate CLSS effectively, such as analysis of nerve root sedimentation signs [23], a new classification according to a hypertrophied ligamentum flavum [24], and cauda equina morphological grading [25]. However, this research only used IAPCSA measurement. Thus, our present results may be limited. Future studies should examine the correlation between the IAPCSA and conventional quantitative radiological criteria for CLSS [26,27,28,29,30,31,32,33,34,35,36,37]. In spite of these limitations, this is the first study to document the association of IAPCSA with CLSS. Our results could be used to improve the quality of diagnosis of CLSS.

## 5. Conclusions

IAPCSA is an adjuvant morphological parameter for the diagnosis of CLSS with an optimal cut-off point of 75.88 mm^2^, a sensitivity of 71.6%, a specificity of 68.6%, and an AUC of 0.78. We believe that this new measurement tool can be used to improve the quality of diagnosis of CLSS.

## Figures and Tables

**Figure 1 jcm-09-00214-f001:**
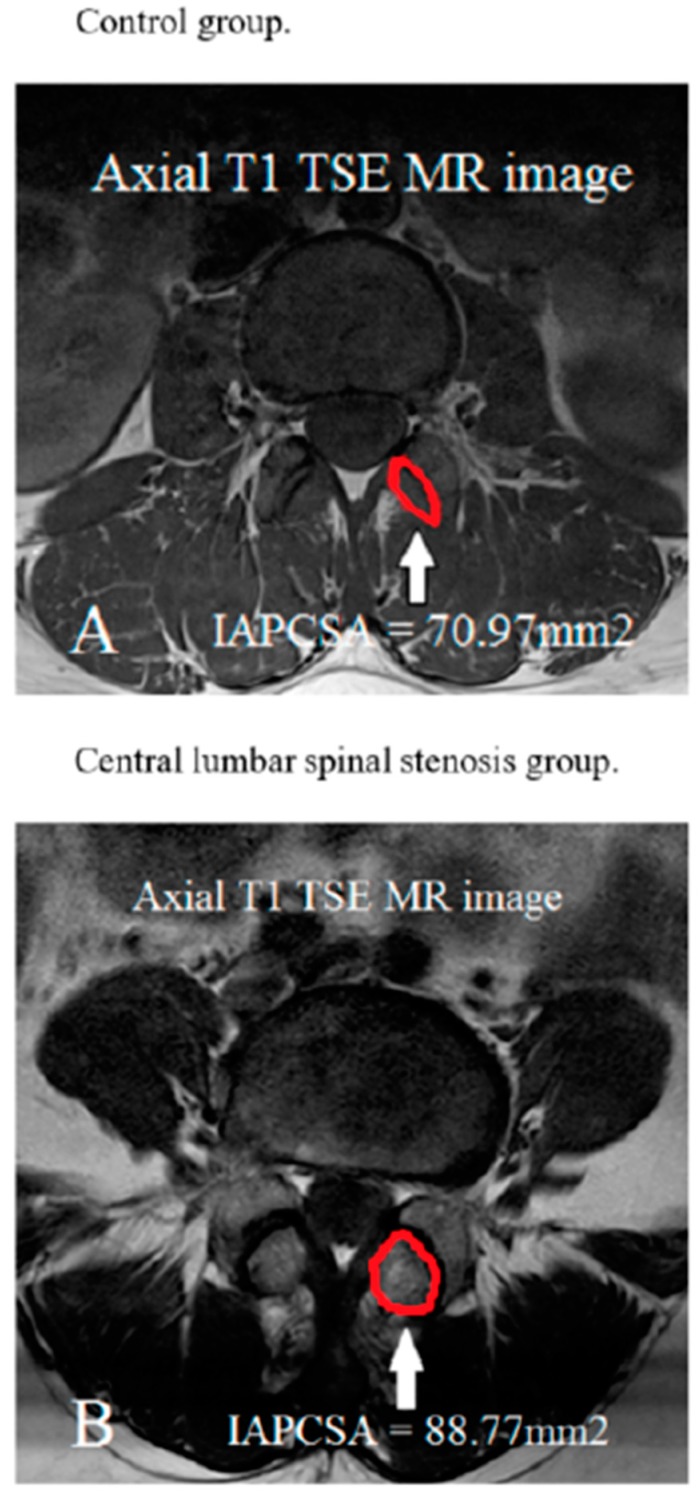
Measurement of the IAPCSA in T1-weighted axial turbo spin echo MRI (TSE MR) at the highest stenotic level.

**Figure 2 jcm-09-00214-f002:**
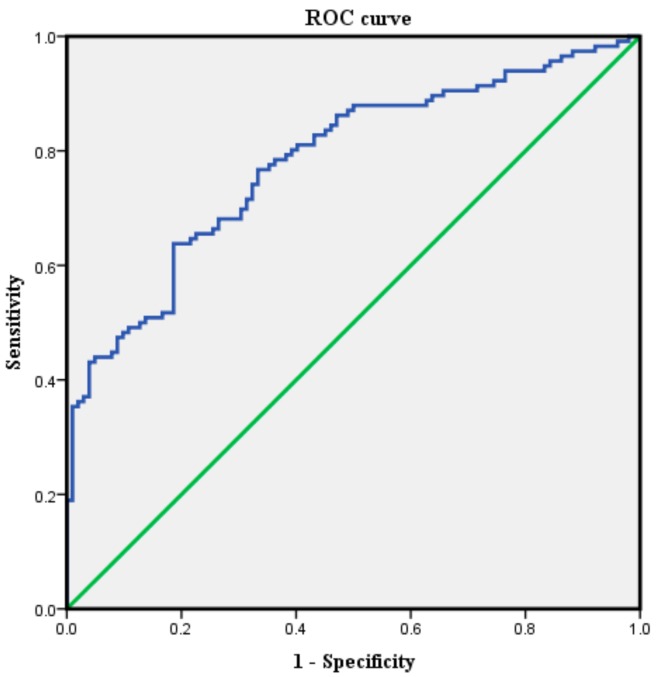
The ROC curve of the inferior articular process’s cross-sectional area for predicting central lumbar spinal stenosis (control group: 102 individuals, central lumbar spinal stenosis group: 116 patients). The best cut-off point was 75.88 mm^2^, with a sensitivity of 71.6%, a specificity of 68.6%, and an AUC of 0.78.

**Table 1 jcm-09-00214-t001:** Comparison of subjects between control and CLSS groups.

Variable	Control Group n = 102	CLSS Group n = 116	Statistical Significance
Gender (male/female)	22/80	38/78	NS
Age (yrs) IAPCSA (mm^2^)	62.25 ± 8.51 70.97 ± 13.02	63.14 ± 8.41 88.77 ± 18.52	NS *p* < 0.001

Data represent the mean ± standard deviation (SD) or the number of patients. NS = not statistically significant (*p* > 0.05). CLSS = central lumbar spinal stenosis; IAPCSA = inferior articular process cross-sectional area.

**Table 2 jcm-09-00214-t002:** The age distribution of patients with the mean IAPCSA in the control group.

Age Distribution (Years)	Total (N)
50–59	69.40 ± 12.31 mm^2^ (46)
60–69	70.28 ± 13.39 mm^2^ (36)
70–82	75.81 ± 13.43 mm^2^ (20) *p* value = 0.172

**Table 3 jcm-09-00214-t003:** The age distribution of patients with the mean IAPCSA of the CLSS group.

Age Distribution (Years)	Total (N)
50–59	91.38 ± 21.36 mm^2^ (44)
60–69	85.93 ± 17.09 mm^2^ (44)
70–83	89.14 ± 15.65 mm^2^ (28) *p* value = 0.386

**Table 4 jcm-09-00214-t004:** Sensitivity and specificity of cut-off value of IAPCSA.

IAPCSA (mm^2^)	Sensitivity (%)	Specificity (%)
45.02	100	2.0
60.74	94.0	20.6
70.98	82.8	55.9
75.88 ^a^	71.6	68.6
84.88	55.2	81.4
98.11	35.3	98.0

^a^ The optimal cut-off value on the receiver operating characteristic (ROC) curve; IAPCSA = inferior articular process cross-sectional area.

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
