# Peer review of "Usefulness of the Inferior Articular Process’s Cross-Sectional Area as a Morphological Parameter for Predicting Central Lumbar Spinal Stenosis"

_jcm, 2020, doi:10.3390/jcm9010214_

Round 1

Reviewer 1 Report

Review of the manuscript entitled “Usefulness of inferior articular process cross sectional area as a morphological parameter for predicting central lumbar spinal stenosis”:

In this study, the authors aimed to determine the usefulness of the inferior articular process cross-sectional area (IAPCSA) for the diagnosis of central lumbar spinal stenosis (CLSS).

To this end, they determined the inferior articular process cross-sectional area (IAPCSA) at the L4/5 level in the MRIs of 102 patients without CLSS and 167 patients who had been clinically diagnosed with CLSS. They found that patients with CLSS on average had a significantly greater IAPCSA. Further, they determined 75.88 mm2 to be the most suitable cut off-point of IAPCSA for predicting CLSS. They concluded that IAPCSA is a useful morphological predictor in the evaluation of CLSS.

This is a fine study about the usefulness of IAPCSA for the diagnosis of CLSS. The paper is generally written in a clear and concise way, however, at some stages it needs clarification. The methods are adequate to address the hypothesis (with minor concerns, see below). The relevant literature is quoted. I would have appreciated, if the study had also examined the correlation between the IAPSCA and conventional quantitative radiological criteria for spinal stenosis (such as described i.E. by Steurer et al. in BMC Musculoskeletal Disorders 2011,  12:175).

Abstract:

L 14     “We hypothesize that the inferior articular process cross-sectional area (IAPCSA) is an important morphological character in the diagnosis of CLSS.”

I think the hypothesis could be stated clearer: i.E.: ....that the inferior articular process cross-sectional area (IAPCSA) is bigger in patients with CLSS compared to those without CLSS.

L 17     “Axial T1-weighted images were obtained from the two subjects.”

“Two subjects” is probably intended as the two groups of 116 patients and 102 controls, please rephrase.

Introduction:

L 30     “Clinical manifestation of CLSS is chronic pain in the waist which affects normal work and daily life and limits activity [2].”

I am not sure whether it is adequate to attribute the pain only to the waist, as patients might also have radicular leg pain. Moreover, there might be better references to illustrate the clinical presentation of CLSS.

Material and methods:

L 51     “Patients’ medical charts were acquired from the musculoskeletal imaging system.”

Please, indicate whether consecutive patients were included, and which was the time frame of the study.

L 57     “A total of 116 patients had CLSS diagnosis which was confirmed by two board-certified, experienced musculoskeletal radiologists…”

Was this diagnosis based on clinical or on morphological criteria?

L 67     “LS-MRI examination was 67 conducted and T1-weighted axial turbo spin echo (TSE) images were acquired at a thickness < 4 mm,….”

Given a slice thickness of 4 mm, in many patients, the inferior articular process might have been visualized in more than one slice. How was the slice selected, in which the IAPCSA was finally calculated? How was investigator bias ruled out at that stage?

L 73     “A PACS  system was used to measure the IAPCSA at the L4/5 facet joint level on lumbar spine MRI.”

Please describe more in detail how the IAPCSA was determined.

Results:

The results section contains some redundant information which is also given in the tables.

Discussion:

L110-119

In this paragraph, there is the great overlap with the beginning of the introduction. This information should be transferred to the introduction. The discussion should then start with the basic findings of the study.

Table 2 and table 3:

Consider integrating both tables to 1 table and also giving P values, comparing the IAPSCA values in the different age groups.

Table 4:

How were the values in the left row determined? How was that ROC-analysis performed?

Figure 2:

I guess, that the labelling of the X-axis should be inverse, beginning with 1 on the left side to 0 on the right side.

Under “authors contributions” it is stated, that YUK and SHL (probably the first author, who in the title page is introduced without a middle name) contributed with funding acquisition. However, no information is given on how the study was funded.

Author Response

We thank again the reviewers for their valuable comments and constructive suggestions. Here we have provided point-to-point responses to the reviewers’ comments. All changes have been highlighted and underlined in the revised manuscript. The reviewers’ critiques are in blue, and our responses are presented in black.

Responses to Reviewer #1

This is a fine study about the usefulness of IAPCSA for the diagnosis of CLSS. The paper is generally written in a clear and concise way.

 Abstract:

L 14     “We hypothesize that the inferior articular process cross-sectional area (IAPCSA) is an important morphological character in the diagnosis of CLSS.”

I think the hypothesis could be stated clearer: i.E.: ....that the inferior articular process cross-sectional area (IAPCSA) is bigger in patients with CLSS compared to those without CLSS.

I appreciate your valuable suggestion.  We revised the manuscript as you recommended.

L 17     “Axial T1-weighted images were obtained from the two subjects.”

“Two subjects” is probably intended as the two groups of 116 patients and 102 controls, please rephrase.

Thank you. You are right.

Introduction:

L 30     “Clinical manifestation of CLSS is chronic pain in the waist which affects normal work and daily life and limits activity [2].”

I am not sure whether it is adequate to attribute the pain only to the waist, as patients might also have radicular leg pain. Moreover, there might be better references to illustrate the clinical presentation of CLSS.

I totally agree with the comment. We revised the manuscript as you pointed out.

Material and methods:

L 51     “Patients’ medical charts were acquired from the musculoskeletal imaging system.”

Please, indicate whether consecutive patients were included, and which was the time frame of the study.

Thanks for your valuable question.

This research was reviewed and approved by the International St Mary`s hospital, Catholic Kwandong University, College of Medicine, Incheon, Institutional Review Board (IRB) (IRB number: IS18RISI0046). We reviewed patients who underwent LS-MRI between June 2017 and April 2019 and had been diagnosed with CLSS.

L 57     “A total of 116 patients had CLSS diagnosis which was confirmed by two board-certified, experienced musculoskeletal radiologists…”

Was this diagnosis based on clinical or on morphological criteria?

Thanks for your valuable question. The CLSS diagnosis  was based on both clinical and morphological criteria.

L 73     “A PACS  system was used to measure the IAPCSA at the L4/5 facet joint level on lumbar spine MRI.” 

Please describe more in detail how the IAPCSA was determined.

I appreciate your valuable suggestion. We revised the manuscript as you pointed out.

Results:

The results section contains some redundant information which is also given in the tables.

I appreciate your valuable suggestion. We revised the manuscript as you pointed out.

Discussion:

L110-119

In this paragraph, there is the great overlap with the beginning of the introduction. This information should be transferred to the introduction. The discussion should then start with the basic findings of the study

I agree with your comment.

The aim of this research is to investigate the relationship between CLSS and IAPCSA using LS-MRI. We demonstrated a positive correlation between CLSS and IAPCSA. This research found that the cut off-point of  IAPCSA at 75.88 mm2 had sensitivity of 71.6%, specificity of 68.6%, and AUC of 0.78 to predict CLSS. Our results suggest that IAPCSA is an accurate and objective morphological parameter for CLSS prediction.

Table 2 and table 3:

Giving P values, comparing the IAPSCA values in the different age groups.

I appreciate your valuable suggestion. We revised the tables as you pointed out.

Table 4:

How were the values in the left row determined? How was that ROC-analysis performed?

Figure 2:

I guess, that the labelling of the X-axis should be inverse, beginning with 1 on the left side to 0 on the right side.

Thanks for your valuable question. The validity of IAPCSA for diagnosis of CLSS was calculated using ROC curves analysis. Statistical analysis was carried out using IBM SPSS version 22 for Windows. And the labelling of the X-axis was right. Thank you so much.

I would have appreciated, if the study had also examined the correlation between the IAPSCA and conventional quantitative radiological criteria for spinal stenosis (such as described i.E. by Steurer et al. in BMC Musculoskeletal Disorders 2011,  12:175).

I appreciate your valuable suggestion. As you pointed out, we have added this sentence in the limitation.

Future studies should examine the correlation between the IAPSCA and conventional quantitative radiological criteria for CLSS.

Under “authors contributions” it is stated, that YUK and SHL (probably the first author, who in the title page is introduced without a middle name) contributed with funding acquisition. However, no information is given on how the study was funded.

Thanks for your valuable question. There is no funding acquisition. It was our mistake. We deleted it. Thank you so much. All authors declare no funding acquisition.

Responses to Reviewer #2

Abstract- Please correct error (or clarify) the sentence in the abstract “Axial T1-weighted images were obtained from the two subjects. In the previous sentence states data from 116 patients with CLSS were obtained as described in Table 3.

I appreciate your valuable suggestion. We revised the manuscript as you recommended.

Figure 2-clarify the number of individuals involved.

I appreciate your valuable suggestion. As you pointed out, we clarified the number of individuals  in the Figure 2.

Discussion-Greater discussion should be included regarding findings from previous research finding involving the superior articular process…instead of a general statement that higher cross-sectional area values are associated with higher probably of CLSS. How similar (or dissimilar) are the cut-off points between superior and inferior articular processes? Perhaps compare and contrast CSA data from superior vs inferior articular process, or at least provide some type of point of reference between the two for readers.

Thanks for your valuable question.

Referenc 10. Lim TH, Choi SI, Cho HR, Kang KN, Rhyu CJ, Chae EY, Lim YS, Lee Y, Kim YU. Optimal Cut-Off Value of the Superior Articular Process Area as a Morphological Parameter to Predict Lumbar Foraminal Stenosis. Pain Res Manag2017;2017:7914836.

In this investigation, the best cut-off point of the superior articular process area was 112.12 mm2 with the sensitivity of 84.4% and the specificity of 83.9%. This value is higher than that of the IAPCSA. The cause of the difference between these two values, is presumably, that the CSA of the superior articular process is larger than that of the IAP.

Minor:

Pg. 6 line 119 Replace “researches” with Previous investigators have reported associations of superior…

I appreciate your valuable suggestion. We revised the manuscript as you recommended.

We would like to cordially inform you that there were some changes in the profile of one of the authors. Currently Dr Sooho Lee is working at Asan Medical Center, Seoul, South Korea. So we changed the profile of Dr Lee in the manuscript.

Thanks again for reviewing our manuscript and giving us your consideration. We hope that our paper is able to meet the Journal of clinical medicine.

Sincerely yours

Reviewer 2 Report

Abstract- Please correct error (or clarify) the sentence in the abstract “Axial T1-weighted images were obtained from the two subjects. In the previous sentence states data from 116 patients with CLSS were obtained as described in Table 3.

Figure 2-clarify the number of individuals involved.

Discussion-Greater discussion should be included regarding findings from previous research finding involving the superior articular process…instead of a general statement that higher cross-sectional area values are associated with higher probably of CLSS. How similar (or dissimilar) are the cut-off points between superior and inferior articular processes? Perhaps compare and contrast CSA data from superior vs inferior articular process, or at least provide some type of point of reference between the two for readers.

Minor:

Pg. 6 line 119 Replace “researches” with Previous investigators have reported associations of superior…

Author Response

(The authors gave the same response as above.)

Round 2

Reviewer 1 Report

In the manuscript has improved a lot in the present version. My previous comments have sufficiently been addressed.